# Arp2/3 complex activity enables nuclear YAP for naïve pluripotency of human embryonic stem cells

**Nathaniel Paul Meyer[1], Tania Singh[1], Matthew L Kutys[1], Todd G Nystul[2], Diane L Barber[1]***

[1]Department of Cell & Tissue Biology, University of California, San Francisco, San Francisco, United States; [2]Departments of Anatomy and OB-GYN/RS, University of California, San Francisco, San Francisco, United States

**Abstract** Our understanding of the transitions of human embryonic stem cells (hESCs) between distinct stages of pluripotency relies predominantly on regulation by transcriptional and epigenetic programs with limited insight on the role of established morphological changes. We report remodeling of the actin cytoskeleton of hESCs as they transition from primed to naïve pluripotency which includes assembly of a ring of contractile actin filaments encapsulating colonies of naïve hESCs. Activity of the Arp2/3 complex is required for formation of the actin ring, to establish uniform cell mechanics within naïve colonies, to promote nuclear translocation of the Hippo pathway effectors YAP and TAZ, and for effective transition to naïve pluripotency. RNA-sequencing analysis confirms that Arp2/3 complex activity regulates Hippo signaling in hESCs, and impaired naïve pluripotency with inhibited Arp2/3 complex activity is rescued by expressing a constitutively active, nuclear-localized YAP-S127A. Moreover, expression of YAP-S127A partially restores the actin filament fence with Arp2/3 complex inhibition, suggesting that actin filament remodeling is both upstream and downstream of YAP activity. These new findings on the cell biology of hESCs reveal a mechanism for cytoskeletal dynamics coordinating cell mechanics to regulate gene expression and facilitate transitions between pluripotency states.

**\*For correspondence:**
diane.barber@ucsf.edu

**Competing interest:** The authors declare that no competing interests exist.

## Editor's evaluation

This important work identifies a mechanism for cytoskeletal dynamics coordinating cell mechanics to regulate gene expression and facilitate transitions between pluripotency states in human embryonic stem cells. The data were collected and analyzed using convincing and validated methodology and can be used as a starting point for studies of the cell biology of embryonic stem cells.

## Introduction

Derivation of clonal pluripotent stem cells (PSCs) from embryos yields cells with a spectrum of pluripotent states, depending on the species, developmental progression of the embryo, and culture conditions. Clonal mouse embryonic stem cells (mESCs) represent a ground state of pluripotency and closely recapitulate the naïve blastocyst from which they are isolated (*Nichols and Smith, 2009*). In contrast, clonal human and other primate PSCs, as conventionally isolated and maintained, are in a primed state of pluripotency and more closely resemble the post-implantation epiblast (*Nakamura et al., 2016*). To study the naïve state of clonal human PSCs, culture conditions have been developed that dedifferentiate primed human embryonic stem cells (hESCs) to a naïve state of pluripotency (*Theunissen et al., 2014*; *Duggal et al., 2015*; *Takashima et al., 2014*; *Szczerbinska et al., 2019*).

Development of culture conditions that convert and sustain a naïve pluripotent state in human PSCs provided an opportunity to study human development before gastrulation (*Rossant and Tam, 2017*).

Such *in vitro* models of naïve pluripotency provided insights into the transcriptomic, epigenetic, and proteomic programs that maintain a functional naïve pluripotency state in stem cells (*Duggal et al., 2015*; *Warrier et al., 2017*; *Theunissen et al., 2016*). We have limited understanding, however, of how established morphological changes during the transition from primed to naive states are regulated and whether morphological changes regulate state transitions. Notably, remodeling of the actin cytoskeleton in response to intracellular signaling and biophysical cues is a predominant determinant for changes in cell morphology as well as for PSC fate and associated gene expression, proliferation, and differentiation (*Naqvi and McNamara, 2020*). Moreover, the actin cytoskeleton coordinates changes in cell shape which are essential for developmental embryogenesis (*Chalut and Paluch, 2016*), and accordingly, mechanoregulation has been studied for roles in exit from the pluripotent state toward targeted cell fates including endodermal (*Chen et al., 2020*), ectodermal (*Keung et al., 2012*), and mesodermal (*Przybyla et al., 2016*) lineages (*Ireland and Simmons, 2015*). Directly targeting actin filament dynamics has been shown to also regulate PSC fate (*Hogrebe et al., 2020*; *Rosowski et al., 2015*; *Gerecht et al., 2007*). Additionally, other components of the cytoskeleton, such as microtubules and intermediate filaments, have been established to modulate stem cell behavior, although studies have primarily focused on how they impact nucleus morphology and activity (*Putra et al., 2023*; *Romero et al., 2022*; *Ndiaye et al., 2022*).

Actin-associated proteins, including β-catenin for enabling Wnt pathway activity, facilitate the maintenance of the naïve pluripotent state in mESCs (*De Belly et al., 2021*). During murine preimplantation development, actin filaments generate mechanical forces that contribute to differentiation throughout the blastocyst stage by modulating mechanosensitive signaling pathways such as Hippo signaling (*Hirate et al., 2015*; *Zenker et al., 2018*). These actin structures allow cells within the developing blastocyst to organize based on contractility, coupling mechanosensing, and fate specification (*Maître et al., 2016*). Despite evidence that morphological changes and actin filament remodeling determine naïve pluripotency during mouse development, their roles in hESC naïve pluripotency remain unclear.

We investigated the role of morphological changes during hESC dedifferentiation to a naïve state of pluripotency and identified the assembly of a ring of contractile actin filaments encapsulating naïve but not primed colonies that is tethered to adherens junctions and decorated with phosphorylated myosin light chain (pMLC) and cortactin. We found that activity of the Arp2/3 complex, an actin filament nucleator, but not formins, which also nucleate actin filaments, is necessary for the formation of the actin ring, naïve cell mechanics, including decreased cell-substrate tensional forces and colony formation, and transition to naïve pluripotency. RNAseq analysis suggested a role for Hippo pathway signaling in Arp2/3 regulated naïve pluripotency, which we confirmed by showing increased nuclear localization of the transcriptional co-activators YAP and TAZ in naïve compared with primed hESCs that is blocked by inhibition of Arp2/3 complex activity. Consistent with these findings, naïve pluripotency, as well as the actin filament ring that is blocked by inhibiting Arp2/3 complex activity, is restored by expressing a nuclear-localized non-phosphorylatable YAP (YAP-S127A), indicating that actin filament remodeling is both upstream and downstream of YAP activity. Our data provide new mechanistic insights into how actin filament dynamics regulates the naïve state of hESCs pluripotency and the integration between actin filament remodeling and pluripotency.

## Results
### Actin filament remodeling as hESCs transition to a naïve state
For morphological analysis of pluripotency states, HUES8 primed hESCs were grown on Matrigel and dedifferentiated to naïve pluripotency using previously reported conditions in an mTeSR-based medium supplemented with MEK (PD0325901) and GSK3 (CHIR99021) inhibitors, the adenylyl cyclase activator forskolin, human leukemia inhibitory factor (LIF), basic fibroblast growth factor (bFGF), and ascorbic acid (*Duggal et al., 2015*; *Qin et al., 2016*). We found that this treatment induced colonies to develop a prominent dome-shape by day 6 of dedifferentiation and increase expression of naïve pluripotency markers DNMT3L, DPPA3, KLF2, and KLF4, as determined by rt-PCR (*Figure 1A and B*), which confirms the transition to a naïve state. Staining fixed cells for actin filaments with

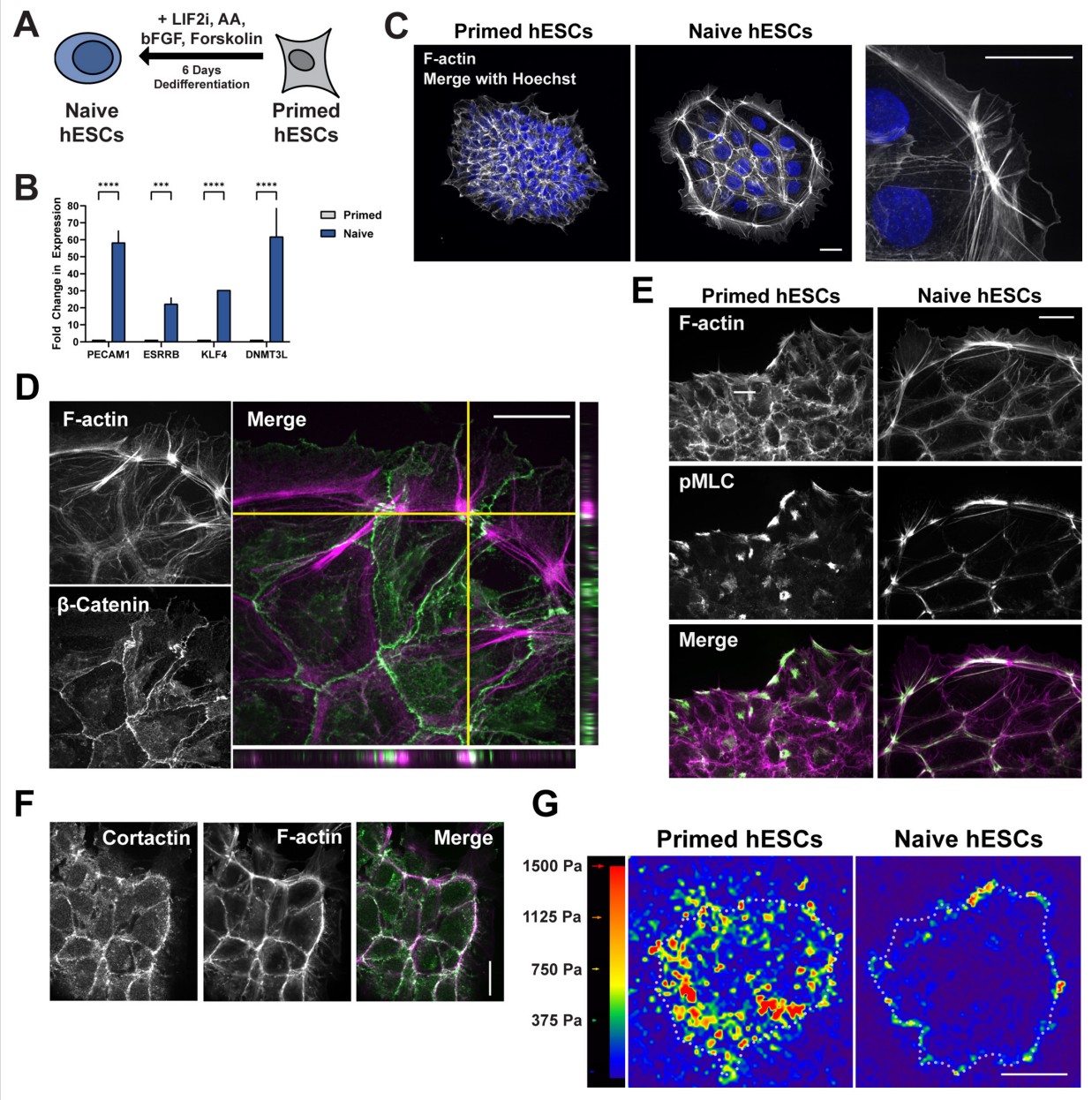

**Figure 1.** Dedifferentiation of primed human embryonic stem cells (hESCs) to naïve pluripotency includes F-actin filament remodeling and the formation of an actin ring. (**A**) Schematic of the dedifferentiation process from primed to naïve hESCs. (**B**) Confirmation of dedifferentiation indicated by increased expression of pluripotency genes associated with a naïve state as determined by quantitative PCR (qPCR). Data represent the means ± SD normalized to Oct4 (n=3 separate cell preparations). Two-way ANOVA with Tukey post hoc test was used to compare between groups. (**C–G**) Images of primed and naïve stem cells stained or immunolabeled for actin cytoskeleton components. (**C**) Confocal (left and middle) and super-resolution (right) images of hESCs stained for F-actin with phalloidin (white) and Hoechst (blue) show a bundled actin filament ring around colonies of naïve but not primed cells. (**D–F**) Confocal images of naïve hESCs immunolabeled for β-catenin (**D**), phosphorylated myosin light chain (pMLC) (**E**), and cortactin (**F**) and stained for F-actin with phalloidin (magenta) to characterize the actin filament ring. Scale bars, 25 µM. (**G**) Representative stress maps generated by traction force microscopy (TFM). Dotted outlines indicate colony boarders. Scale bar, 50 µM.

The online version of this article includes the following source data and figure supplement(s) for figure 1:

**Source data 1.** Raw numerical data for *Figure 1B*.

**Figure supplement 1.** The actin ring architecture forms independent of cell line, dedifferentiation media, and colony size.

phalloidin revealed that naïve but not primed colonies had a ring of bundled actin filaments at the colony periphery (*Figure 1C*). A similar actin ring also formed around colonies of naïve H9 cells and naïve WTC11 induced PSCs (iPSCs) (*Figure 1—figure supplement 1A*) as well as HUES8 cells dedifferentiated by alternative medium supplements (*Figure 1—figure supplement 1B*). Moreover, the actin ring assembled independently of naïve colony size (*Figure 1—figure supplement 1C*).

An actin filament ring is reported to encircle colonies of clonal human PSCs to provide a mechanosensitive element linked to focal adhesions (*Närvä et al., 2017*). The actin filament ring we observed around naïve hESC colonies was instead tethered to adherens junctions, as indicated by co-labeling for β-catenin, with separated interdigitated adherens junctions suggesting a contractile force (*Figure 1D*, crosshairs). The contractile property of the ring was also suggested by the actin ring around naïve hESC colonies being decorated with pMLC as determined by immunolabeling (*Figure 1E*). In contrast, primed hESC colonies had irregular aggregates of pMLC with limited co-localization with actin filaments. Additionally, immunolabeling for cortactin, a regulator of cortical branched actin filaments, partly overlapped with the actin ring (*Figure 1F*). The subcellular localization of the actin ring was confirmed with 3D reconstruction (*Figure 1—figure supplement 1D*). Together, these data indicate a contractile actin ring surrounding naïve but not primed hESC colonies, with the ring likely composed of unbranched actin filaments, which bind pMLC, and branched filaments, which bind cortactin.

The nature of the actin ring enclosing colonies of naïve but not primed hESC colonies suggested a potential difference in colony mechanics, which we determined by using traction force microscopy. Increased cell-matrix traction forces are associated with destabilized adherens junctions in epithelial monolayers (*Mertz et al., 2013*; *Scarpa et al., 2015*). Consistent with pMLC localization, primed colonies exhibited elevated cell-substrate tractions that were distributed throughout the colony (*Figure 1G*, left; *Figure 1—figure supplement 1E*). In contrast, naïve colonies exhibited overall low magnitude cell-substrate tractions that were localized to the colony periphery and largely absent from the colony interior (*Figure 1G*, right; *Figure 1—figure supplement 1E*), suggesting decreased cell-substrate tensional force and a likely shift to more stabilized cell-cell forces. Along with pMLC localization, these low traction forces are consistent with uniform cell-cell adhesion in naïve hESC colonies. Together these data reveal a significant reorganization of the actin cytoskeleton during the transition to a naïve state of pluripotency that includes the assembly of a contractile actin ring surrounding naïve cell colonies, coincident with attenuated cell-substrate traction forces and a transition to enhanced cell-cell junction traction force within the colony unit.

## Arp2/3 complex activity is necessary for transition of hESC to naïve pluripotency

The assembly of an actin ring in naïve but not primed hESC colonies led us to ask whether the actin ring has a functional significance in the transition to naïve pluripotency. New actin filaments are predominantly generated by two distinct nucleators, the Arp2/3 complex, which generates branched filaments, and formins, which generate unbranched filaments (*Pollard, 2007*). We found that the actin ring assembled when naïve cells are generated in the presence of SMIFH2, a broad-spectrum inhibitor of formin activity (*Rizvi et al., 2009*; *Ganguly et al., 2015*) but not CK666, a selective pharmacological inhibitor of Arp2/3 complex activity (*Nolen et al., 2009*; *Yang et al., 2012* ; *Figure 2A*), despite the ring likely being composed of both unbranched and branched actin filaments as indicated by pMLC and cortactin immunolabeling. Additionally, CK666 blocked increased expression of markers of naïve pluripotency seen in controls, determined by quantitative PCR (qPCR) of PECAM1, ESRRB, KLF4, and DNMT3L (*Figure 2B*). To eliminate the possibility that CK666 treatment led cells to exit pluripotency and differentiate, we immunolabeled for the general pluripotency markers OCT4 and SOX2 and found that CK666-treated cells remained broadly pluripotent (*Figure 2—figure supplement 1A and B*). To further confirm that Arp2/3 complex activity is necessary for the actin filament ring, we also treated cells with CK869, another inhibitor of Arp2/3 complex activity (*Hetrick et al., 2013*) . Consistent with our finding using CK666, CK869 blocked the formation of the actin ring without impairing pluripotency (*Figure 2—figure supplement 1A and B*).

To further assess the pluripotent state of cells dedifferentiated in the presence of CK666, we immunolabeled for the primed pluripotent marker SSEA3 (*Trusler et al., 2018*). SSEA3 expression significantly decreased with dedifferentiation in control conditions, as previously reported (*Liu et al., 2017*) but remained at levels similar to primed cells in the presence of CK666 (*Figure 2C and D*).

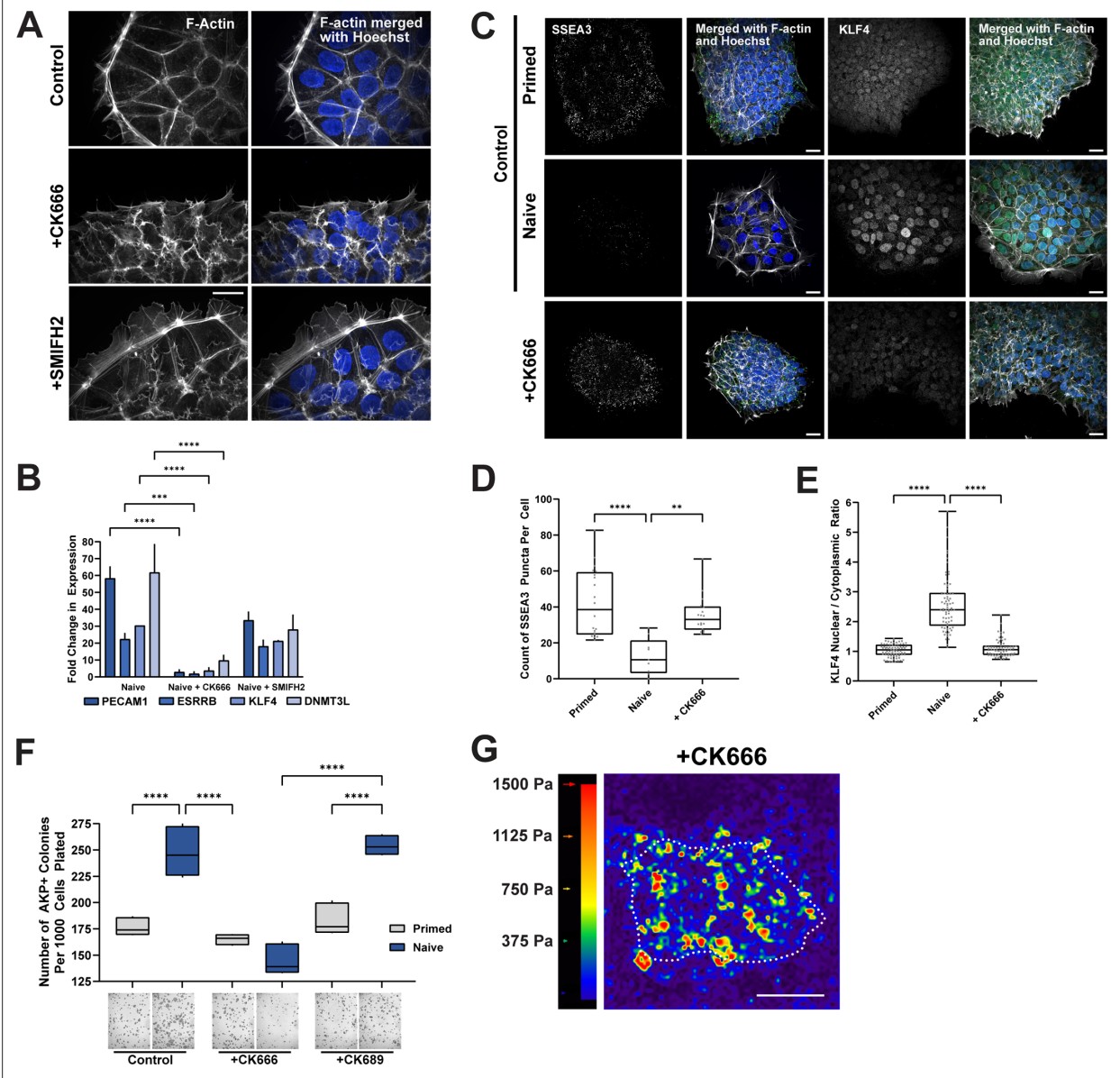

**Figure 2.** Inhibiting Arp2/3 complex but not formin activity blocks formation of an actin ring and dedifferentiation to naïve pluripotency. (**A**) Confocal images of D6 naïve human embryonic stem cells (hESCs) maintained in the absence (Control) or presence of 80 µM CK666 or 50 µM SMIFH2 and stained for F-actin with phalloidin and nuclei with Hoechst. (**B**) Expression of the indicated pluripotency transcripts determined by quantitative PCR (qPCR) at D6 of dedifferentiation in the absence (Control) or presence of CK666 or SMIFH2. The Arp2/3 complex activity inhibitor CK666 impairs upregulation of pluripotency genes used to identify naïve pluripotency. Data are the means ± SD of three determinations normalized to Oct4. Two-way ANOVA with Tukey post hoc test was used to compare between groups. (**C–E**) Confocal images of control primed and naïve hESCs and D6 cells dedifferentiated in the presence of CK666 immunolabeled for the primed marker SSEA3, quantified in (**D**) and the naïve marker KLF4, quantified in (**E**). Box plots in (**D**) and (**E**) show median, first and third quartile, with whiskers extending to observations within 1.5 times the interquartile range. (**F**) Clonogenicity, determined by alkaline phosphatase positive colonies (quantified in top panel and representative brightfield images in bottom panel) in control primed and naïve hESC as well as dedifferentiated in the presence of CK666 or 80 µM CK689, an inactive analog of CK666. Data are the means ± SD normalized to the number of cells plated in three separate determinations. Box plots are as described in (**D, E**) and one-way ANOVA with Tukey post hoc test was used to compare between groups. Scale bars, 25 µM. (**G**) Representative stress maps generated by traction force microscopy (TFM). Dotted outlines indicate colony boarders. Scale bar, 50 µM.

The online version of this article includes the following source data and figure supplement(s) for figure 2:

**Source data 1.** Raw numerical data for *Figure 2B, D, E, and F*.

**Figure supplement 1.** Inhibiting Arp2/3 complex activity in hESCs does not lead to an exit from pluripotency and does not alter rates of proliferation.

**Figure supplement 1—source data 1.** Raw numerical data for *Figure 2—figure supplement 1B and C*.

Additionally, the naïve pluripotency marker KLF4 (*Takashima et al., 2014*) translocated from the cytoplasm to the nucleus with control dedifferentiation but not in the presence of CK666 (*Figure 2C and E*). Although branched actin filaments generated by Arp2/3 complex promote cell cycle progression and proliferation (*Molinie et al., 2019*), we used an EdU pulse (1 hour) to show no difference in EdU incorporation in the absence or presence of CK666 or CK869 (*Figure 2—figure supplement 1C*), indicating that the inhibitors did not change cell proliferation. Consistent with CK666 disrupting the actin filament ring, it also disrupted localization of cortactin around naïve colonies (*Figure 2—figure supplement 1D*).

We further tested effects of CK666 on a functional naïve pluripotent state by staining for alkaline phosphatase and scoring for colony formation, which indicates the capacity for clonogenic expansion and self-renewal (*Rostovskaya et al., 2019*). Primed and naïve hESCs were passaged and plated at clonogenic cell numbers and maintained for 5 days without or with CK666. In controls, colony formation was greater in naïve compared with primed hESC, as previously reported (*Chen et al., 2022*). However, with CK666 but not CK689, an inactive analog of CK666, there was no increase in colony formation in naïve compared with primed cells (*Figure 2F*). Additionally, traction force microscopy revealed that elevated cell-substrate tractions throughout colonies of primed but not naïve cells (*Figure 1G*) were retained when hESCs were dedifferentiated in the presence of CK666 (*Figure 2G*; *Figure 2—figure supplement 1E*). These data identify an essential role for the Arp2/3 complex in promoting an actin filament ring and uniform naïve colony mechanics as well as acquiring a naïve pluripotent state in hESCs.

## Arp2/3 complex activity enables active YAP for naïve pluripotency

To understand how Arp2/3 complex activity affects the transcriptional circuitry required for naïve pluripotency, we performed bulk RNA-sequencing (RNAseq) on primed and naïve hESCs and hESCs dedifferentiated in the presence of CK666 (*Figure 3A*). We found that primed, naïve, and CK666-treated cells had a total of 12,817 differentially expressed genes (DEGs) with an adjusted pval<0.05 (false discovery rate [FDR]-corrected by Benjamini-Hochberg procedure). Of these DEGs, 182 were unique to control primed cells compared with control naïve cells and were not differentially expressed in CK666-treated cells; CK666-treated cells compared with control primed or control naïve cells had 102 and 502 DEGs, respectively. To determine the transcriptional networks involved in the dedifferentiation from primed to naïve pluripotency, we identified KEGG pathways in control naïve dedifferentiation which revealed Hippo signaling as the top candidate (*Figure 3B*). Additionally, transcription factor binding motif analysis revealed that one of the top candidates was TEAD2 (*Figure 3C*), which is a downstream effector of Hippo signaling.

The Hippo effector protein YAP is a known regulator of the human naïve pluripotent state, with overexpression of YAP in PSCs promoting the acquisition of naïve pluripotency (*Qin et al., 2016*). Although actin filament dynamics is reported to regulate YAP signaling (*Hsiao et al., 2016*; *Furukawa et al., 2017*), to our knowledge a role for Arp2/3 complex activity regulating YAP or TAZ activity in human naïve pluripotency has not been reported. For an unbiased global analysis of known YAP target genes, we used two publicly available datasets (*Estarás et al., 2017*; *Pagliari et al., 2021*) and found that of the 3744 YAP target genes identified in our RNAseq dataset, 3156 (84%) were not differentially expressed in any condition and 588 (16%) were enriched in one or multiple conditions. Of those 588 enriched YAP target genes, 174 (30%) were significantly enriched in the control naïve dedifferentiation condition versus the control primed condition; 407 (69%) were significantly enriched among CK666-treated dedifferentiation condition versus the control primed condition; and 7 (1%) were significantly enriched in both conditions versus the control primed condition (*Figure 3D*, adjusted pval>0.05).

Of the genes significantly enriched in the control naïve condition compared with the control primed condition, known naïve pluripotency markers such as OTX2, DLG2, and CRY1 were significantly upregulated, and these naïve markers were not significantly increased in CK666-treated cells (*Figure 3E*, left). As expected, genes significantly enriched among both DEG lists included known YAP and Hippo targets such as ANKRD1, SLIT2, and CHD10 (*Figure 3E*, right). Genes significantly enriched among the CK666-treated condition include the negative Hippo regulator AMOT (*Zhao et al., 2011*), and lineage-commitment genes such as SOX6 and SPEF2 (*Figure 3E*, middle). To verify this prediction, we immunolabeled cells to determine YAP localization and found increased nuclear to cytoplasmic ratios of YAP (*Figure 3F and G*) and TAZ (*Figure 3—figure supplement 1A and B*) with control

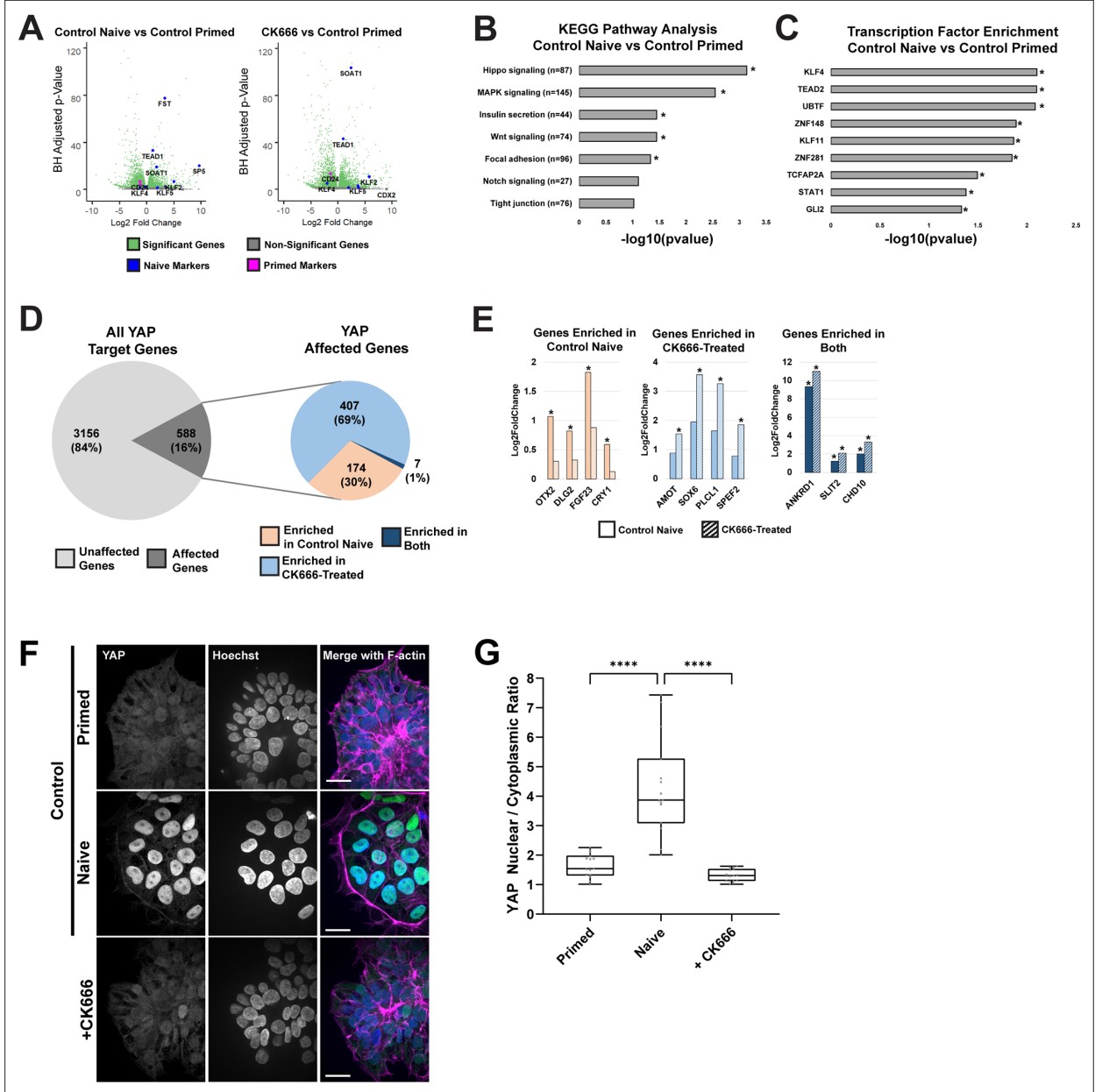

**Figure 3.** Inhibiting Arp2/3 complex activity disrupts Hippo signaling in naïve human embryonic stem cells (hESCs). (**A**) Volcano plots showing transcriptome fold-changes (padj) of dedifferentiations in the absence (Control Naïve) or presence of CK666-treated dedifferentiations compared with primed hESCs. Each dot represents a single gene, with significant genes (padj<0.05; false discovery rate [FDR]-corrected by Benjamini-Hochberg procedure) in green. Notable primed and naïve markers are depicted in magenta and blue, respectively. (**B, C**) KEGG pathway analysis (**B**) and transcription factor enrichment analysis (**C**) of control primed and naïve hESCs. The number of differentially expressed genes (DEGs) indicated in each pathway is displayed and asterisks indicate significantly enriched pathways (p<0.05). (**D**) Unbiased screening of all known YAP target genes in dedifferentiated cells in the absence (Control Naive) and presence of CK666. Affected genes were further analyzed to indicate whether they are enriched in control, CK666-treated, or both conditions when compared with primed controls. (**E**) Expression of selected YAP target genes from bulk RNA-sequencing (RNAseq), with asterisks indicating significant difference (padj<0.05). (**F**) Representative confocal images of control primed and naïve hESCs and naïve hESCs generated in the presence of 80 µM CK666 immunolabeled for YAP and stained for nuclei with Hoechst and F-actin with phalloidin. (**G**) Quantification of nuclear to cytoplasmic ratio of YAP from images as shown in (**F**). Box plots show median, first and third quartile, with whiskers extending to observations within 1.5 times the interquartile range. Data are from five separate cell preparations and one-way ANOVA with Tukey post hoc test was used to compare between groups. Scale bars, 25 µM.

The online version of this article includes the following source data and figure supplement(s) for figure 3:

**Source data 1.** Raw numerical data for *Figure 3G*.

*Figure 3 continued on next page*

*Figure 3 continued*

**Figure supplement 1.** Inhibiting Arp2/3 complex activity disrupts TAZ localization in naïve hESCs.

**Figure supplement 1—source data 1.** Uncropped and labeled gels for *Figure 3—figure supplement 1C and D*.

**Figure supplement 1—source data 2.** Raw unedited gels for *Figure 3—figure supplement 1C and D* .

**Figure supplement 1—source data 3.** Raw numerical data for *Figure 3—figure supplement 1B and D*.

dedifferentiation that was blocked by CK666. Immunoblotting total cell lysates for total YAP abundance indicated no difference between primed and naïve hESCs and a small but significant increase in hESCs treated with CK666 (*Figure 3—figure supplement 1C and D*). These data indicate that a Hippo signaling pathway program, driven by mediators such as YAP, is active during dedifferentiation to naïve pluripotency but is disrupted by inhibiting Arp2/3 complex activity.

Consistent with our findings, actin filaments and associated proteins generate mechanical forces during preimplantation development that contribute to differentiation throughout the blastocyst stage by modulating mechanosensitive pathways such as Hippo (*Hirate et al., 2015*; *Zenker et al., 2018*). These actin structures allow cells within the developing blastocyst to organize based on contractility, coupling mechanosensing, and fate specification (*Maître et al., 2016*). Therefore, we hypothesized that Arp2/3 complex activity facilitated naïve dedifferentiation through increasing YAP nuclear localization. To test this prediction, we asked whether primed hESCs stably expressing a constitutively active, nuclear-localized YAP (YAP-S127A) could restore naïve dedifferentiation in the presence of CK666. Accordingly, we found that that expression of YAP-S127A in the presence of CK666 restored two markers of the naïve state, increased nuclear localization of KLF4 (*Figure 4A and B*) and decreased SSEA3 (*Figure 4C and D*). In contrast, acquisition of a naïve pluripotency state remained blocked with CK666 treatment in cells overexpressing wildtype YAP (YAP-WT) (*Figure 4A–D*). Colony formation, a functional form of naïve pluripotency, was also restored by heterologous expression of YAP-S127A but not YAP-WT in the presence of CK666 (*Figure 4E*), indicating that Arp2/3 complex activity is necessary for active nuclear-localized YAP to induce a naïve pluripotency state. In addition, we found that expressing YAP-S127A but not YAP-WT in the presence of CK666 partially restored a contractile actin ring enclosing naïve cell colonies (*Figure 4F*), indicating bidirectional signaling between actin filament remodeling and active nuclear-localized YAP for assembly of the actin filament ring around naïve hESC colonies. Thus, we conclude that both the transition from primed to naïve hESC pluripotency includes an Arp2/3 complex-dependent actin filament remodeling that enables active nuclear-localized YAP, and that nuclear-localized YAP enables actin filament remodeling for naïve pluripotency.

## Discussion

Our new findings support a model in which naïve pluripotency is characterized by an Arp2/3 complex-dependent remodeling of the actin cytoskeleton that includes formation of a contractile actin ring enclosing naïve colonies and establishment of uniform tensional forces in colonies likely enabled by the actin ring being physically associated with β-catenin and pMLC, which are known to play roles in pluripotency (*Xu et al., 2016*; *De Belly et al., 2021*). Moreover, Arp2/3 activity facilitates dedifferentiation to a naïve state of pluripotency through promoting nuclear translocation of YAP and regulating Hippo target gene expression. Consistent with these findings, whereas the transition to naïve pluripotency is blocked with inhibiting Arp2/3 complex activity, this transition is restored by expression of a constitutively active, nuclear-localized YAP-S127A.

The contractile rings we describe are distinct from those that assemble around *Xenopus* neural crest cells (*Shellard et al., 2018*), which function to enhance migratory capacity, and around colonies of iPSCs (*Närvä et al., 2017*), which function to enhance cell-substrate adhesion. Induced pluripotency has long been proposed to be closer to naïve pluripotency than primed stem cells as conventionally isolated and maintained (*Nichols and Smith, 2009*; *Theunissen et al., 2016*). Our findings also highlight distinct differences between murine cells and hESCs. Cells within the ICM of mouse blastocysts exclude YAP from the nucleus whereas cells within the ICM of human blastocysts maintain nuclear YAP (*Qin et al., 2016*; *Nishioka et al., 2009*). This difference in YAP localization is retained *in vitro*, with murine naïve PSCs having predominantly cytosolic YAP (*Chung et al., 2016*), and human naïve PSCs having predominantly nuclear YAP (*Figure 3F and G*). How this difference in YAP localization occurs

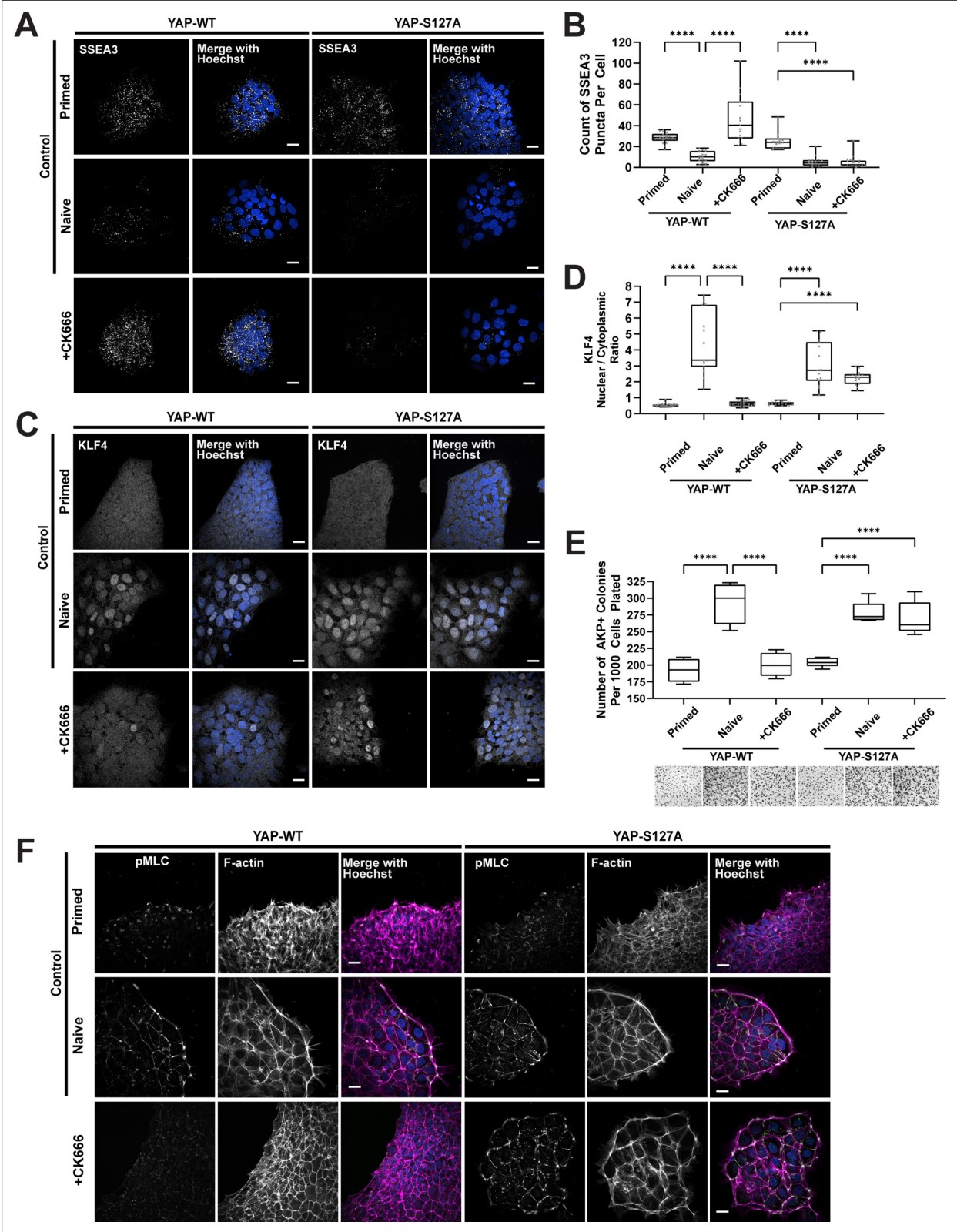

**Figure 4.** Overexpression of YAP-S127A rescues naïve pluripotency blocked with inhibiting Arp2/3 complex activity. (**A, C**) Representative confocal images of control primed, control naïve cells, and cells dedifferentiated in the presence of CK666 with or without stably overexpressing YAP-WT or YAP-S127A immunolabeled for the primed marker SSEA3 (**A**) or the naïve marker KLF4 (**C**) and stained for nuclei with Hoechst and F-actin with phalloidin. (**B, D**) Images as in (**A**) and (**C**) were used to quantify, respectively, the number of SSEA3 puncta (**B**) and the nuclear to cytoplasmic ratio of KLF4 (**D**). Box

*Figure 4 continued on next page*

*Figure 4 continued*

are plots as described for *Figure 2*. (**E**) Clonogenicity, determined by alkaline phosphatase positive colonies (quantified in top panel and representative brightfield images in bottom panel) in control primed and naïve human embryonic stem cell (hESC), and dedifferentiated in the presence of CK666 with stably expressed YAP WT or YAP-S127A. Data are the means ± SD normalized to the number of cells plated from five separate determinations, with box plots as described for *Figure 2d and e* and one-way ANOVA with Tukey post hoc test was used to compare between groups. Representative confocal images of control primed, control naïve cells, and cells dedifferentiated in the presence of CK666 with or without stably expressing YAP-WT or YAP-S127A immunolabeled for phosphorylated myosin light chain (pMLC) and stained for nuclei with Hoechst and F-actin with phalloidin. Scale bars, 25 μM.

The online version of this article includes the following source data for figure 4:

**Source data 1.** Raw numerical data for *Figure 4B, D, and E*.

between mouse and human is unknown, although YAP localization is reported to be regulated by stability of the actin cytoskeleton and contractility, as well as various mechanical regulators (*Furukawa et al., 2017*).

The role of the actin cytoskeleton in exit from the pluripotent state has also highlighted how actin dynamics may facilitate cell fate decisions. For example, cells located at the colony edge of primed hESCs have distinct cytoskeletal dynamics and are uniquely poised to exit pluripotency and differentiate (*Rosowski et al., 2015*; *Kim et al., 2022*). Positional differences in differentiation potential such as these have been proposed as a mechanism executed in early embryo symmetry breaking with rearrangement of the actin cytoskeleton being required for the first cell fate decision in the blastocyst (*Skamagki et al., 2013*; *Chen et al., 2018*). Thus, it may be possible that the contractile actin ring we observe at the edge of naïve colonies (*Figure 1C*) functions as a hub for regulating cell fate dynamics through similar mechanisms as first cell fate decision including modulation of mechanosensitive signaling such as YAP and through pathways such as Hippo. Our finding that the actin ring, which is lost in the presence of CK666, is restored by YAP-S127A but not YAP-WT suggests a complex circuitry between YAP activity and actin filament remodeling.

Further highlighting differences between hESCs and mESCs, we recently reported that Arp2/3 complex activity is necessary for the differentiation of clonal mouse naïve PSCs to the primed epiblast state, which is in part mediated by translocation of myocardin-related transcription factor MRTF from the cytosol to the nucleus (*Aloisio and Barber, 2022*). Additionally, a recent report suggests that Arp2/3 complex activity may form a positive feedback loop with YAP-TEAD1 transcriptional activity controlling cytoskeletal reorganization (*Pagliari et al., 2021*). Thus, Arp2/3 complex activity may regulate naïve pluripotency at multiple stages including initially to reorganize the actin cytoskeleton, but also during maintenance of naïve pluripotency through regulating YAP localization and hence activity.

Taken together, our findings increase our understanding of actin dynamics and cell mechanics as regulators of cell fate transitions. A role for contractile actin filaments as a mechanoresponsive element for pluripotency states is well established (*De Belly et al., 2022*), and our work identifies cytoskeletal dynamics essential for uniform colony mechanics and the naïve pluripotent state, the role of Arp2/3 complex activity, and YAP/TAZ activity as a promising target for reprogramming of hESCs and for regenerative medicine.

## Methods
### Cell culture
Primed hESC lines HUES8 (RRID: CVCL_B207), H9 (RRID: CVCL_1240), and WTC11 (RRID: CVCL_Y803) were maintained on Matrigel (Corning Life Science #354277) in feeder-free mTeSR-1 medium (STEMCELL Technologies # 85850) at 37°C with 5% $CO_2$ with daily medium changes. Cells were passaged approximately every 3 days, by dissociating with Accutase (STEMCELL Technologies #07920) and including the Rho-associated coiled-coil kinase (ROCKi) inhibitor Y-276932 (10 μM; Selleckchem #S1049) in the plating medium to facilitate survival. All cell lines were routinely confirmed to be negative for mycoplasma by testing with a MycoAlert Mycoplasma Detection Kit (Lonza # LT07-701). The HUES8 cell line was authenticated using STR Profiling (Human Cell STR Profiling Service; ATCC).

### Generation of naïve hESCs
Dedifferentiation was completed using previously published methods (*Duggal et al., 2015*; *Qin et al., 2016*). In brief, cells were plated at a density of 10,000 cells per cm (*Nakamura et al., 2016*) in the

presence of ROCKi (10 µM). After 24 hr, cells were washed three times with PBS and incubated in naïve dedifferentiation medium of mTeSR-1 supplemented with 12 ng/mL bFGF (Peprotech #AF-100-18B), 1 µM PD0325901 (MEKi, Selleckchem #S1036), 3 µM CHIR99021 (GSK3βi, Selleckchem #S2924), 10 µM forskolin (STEMCELL Technologies #72112), 50 ng/mL ascorbic acid (Sigma # A92902), and 1000 U recombinant human LIF (STEMCELL Technologies #78055). Medium was replaced daily, and cells were passaged every 3 days with Accutase. Where indicated, the naïve dedifferentiation medium 2iFL was used, which consisted of mTeSR-1 supplemented with 0.5 µM PD0325901, 3 µM CHIR9902, 10 µM forskolin, and 1000 U recombinant human LIF. For actin nucleator experiments, naïve dedifferentiation media was supplemented with either 80 µM CK666 (EMD Millipore #182515), 80 µM CK689 (EMD Millipore #182517), or 50 µM SMIFH2 (Sigma #S4826) throughout the entire dedifferentiation process. When passaging, media was supplemented with both ROCKi (10 µM) and the appropriate inhibitor.

## qPCR

Total RNA was isolated using RNAeasy Mini Plus (QIAGEN #74134) kits and cDNA was generated using iScript cDNA Synthesis kits (Bio-Rad #1708890) as per the manufacturer's specifications. Quantitative PCR was performed using iQ SYBR Green Supermix (Bio-Rad #1708882) and analyzed on a QuantStudio six Flex Real-Time PCR System (Applied Biosystems).

| qPCR primer name | Sequence (5' to 3') |
| --- | --- |
| GAPDH_for | ACAACTTTGGTATCGTGGAAGG |
| GAPDH_rev | GCCATCACGCCACAGTTTC |
| Oct4_for | GTGTTCAGCCAAAAGACCATCT |
| Oct4_rev | GGCCTGCATGAGGGTTTCT |
| Dnmt3l_for | TGAACAAGGAAGACCTGGACG |
| Dnmt3l_rev | CAGTGCCTGCTCCTTATGGCT |
| Klf2_for | ACCAAGAGCTCGCACCTAAA |
| Klf2_rev | GTGGCACTGAAAGGGTCTGT |
| Klf4_for | CGGACATCAACGACGTGAG |
| Klf4_rev | GACGCCTTCAGCACGAACT |
| DPPA3_for | TAGCGAATCTGTTTCCCCTCT |
| DPPA3_rev | CTGCTGTAAAGCCACTCATCTT |
| PECAM1_for | AACAGTGTTGACATGAAGAGCC |
| PECAM1_rev | TGTAAAACAGCACGTCATCCTT |
| ESRRB_for | ATCAAGTGCGAGTACATGCTC |
| ESRRB_rev | CGCCTCCGTTTGGTGATCTC |

## Staining and immunolabeling

For microscopy, cells were plated on Matrigel-coated glass coverslips prepared using an ultrasonic cleaning bath (Branson). In brief, coverslips were sonicated for 20 min in the presence of diluted Versa-Clear (Fisher Scientific #18-200-700) in double distilled $H_2O$ (ddH$_2$O), washed three times using ddH$_2$O, sonicated for 20 min in ddH$_2$O, washed three times using ddH$_2$O, and sterilized and stored in 70% ethanol until use. Cells were maintained for indicated times, typically 3 days, washed briefly with PBS, fixed with 4% PFA for 12 min at room temperature (RT), permeabilized with 0.1% Triton X-100 in PBS for 5 min, and incubated with blocking buffer consisting of 0.1% Triton X-100 in PBS and 1% BSA for 1 hr. Cells were then incubated with primary antibodies diluted in blocking buffer overnight at 4°C, washed with PBS three times, and incubated for 1 hr at RT with secondary antibodies, followed by a final PBS 3× wash, with the second wash containing Hoechst 33342 (1:10,000; Molecular Probes #H-3570) to stain nuclei. To stain for actin filaments, either rhodamine phalloidin (1:400,

Invitrogen #R415) or Phalloidin-iFluor 647 (1:1000, Abcam #ab176759) was added to the secondary antibody incubation. Secondary antibodies used were Alexa Fluor 488 or 594 for the appropriate species primaries (1:500, Invitrogen #A-11037). To assess proliferation, EdU staining was performed according to the manufacturer's guidelines with a 1 hr EdU pulse (Click-iT EdU Cell Proliferation Kit for Imaging, Alexa Fluor 647 dye, Fisher Scientific, #C10340).

| Antibody | Source | Catalog number | RRID | Dilution |
|---|---|---|---|---|
| β-Catenin | BD Transduction | #610154 | AB_397555 | 1:200 ICC |
| pMLC (Thr18/Ser19) | Cell Signaling | #3674 | AB_2147464 | 1:200 ICC |
| pan-ERM | Cell Signaling | #3142 | AB_2100313 | 1:400 ICC |
| Moesin | Cell Signaling | #3146 | AB_2251034 | 1:400 ICC |
| Ezrin | Cell Signaling | #3145 | AB_2100309 | 1:400 ICC |
| SSEA3 | Santa Cruz | sc-21703 | AB_628288 | 1:200 ICC |
| KLF4 | Cell Signaling | #4038 | AB_2265207 | 1:200 ICC |
| YAP | Sigma-Aldrich | HPA038885 | AB_2676255 | 1:400 ICC |
| YAP | Cell Signaling | #4912 | AB_2218911 | 1:1000 WB |
| TAZ | Sigma-Aldrich | HPA039557 | AB_10672899 | 1:400 ICC |
| Oct3/4 (C-10) | Santa Cruz | sc-5279 | AB_628051 | 1:400 ICC |
| Sox2 | Cell Signaling | #3579 | AB_2195767 | 1:400 ICC |
| Cortactin | Cell Signaling | #3503 | AB_2115160 | 1:200 ICC |

## Confocal and super-resolution image acquisition and quantification

Cells were imaged using an inverted microscope system (Nikon Eclipse TE2000 Perfect Focus System; Nikon Instruments) equipped with a spinning-disk confocal scanner unit (CSU10; Yokogawa), a 488 nm solid-state laser (LMM5; Spectral Applied Research), and a multipoint stage (MS-2000; Applied Scientific Instruments). A CoolSnap HQ2 cooled charge-coupled camera (Photometrics) was used to take images with a camera triggered electronic shutter controlled by NIS Elements Imaging Software (Nikon) and a 60× Plan Apochromat TIRF 1.45 NA oil immersion objective equipped with a Borealis (Andor) to normalize illumination. High-resolution and super-resolution images were acquired using a Yokogawa CSU-W1/SoRa spinning disk confocal system (Yokogawa) and an ORCA Fusion BT sCMOS camera (Hamamatsu) using 2×2 camera binning. Nuclear-to-cytoplasmic ratios of immunolabeled proteins and number of puncta per cell were quantified using NIS Elements Imaging Software (Nikon). Briefly, the fluorescence in the nucleus (detected by Hoescht) and in the cytoplasm were manually sampled by selection of regions-of-interest. Three regions-of-interest outside of any cell were used to calculate background fluorescence and was subtracted from both nuclear and cytoplasmic fluorescence values. The ratio of fluorescence was then determined by diving the nuclear fluorescence intensity with that of the cytoplasm for a given cell. Quantification of puncta for SSEA3 was done by creating a 3D projection of full-cell z-stacks by using NIS Elements Imaging Software. Surfaces were created using the 3D thresholding tool normalized across all images and the total number of puncta was recorded. The total number of cells was then counted, as determined by the number of Hoescht positive nuclei, and the number of puncta per cell was calculated by dividing the total number of puncta by the number of cells in each field of view. Percentage of positive cells for Oct4/Sox2 staining and EdU incorporation was determined by manually counting the total number of nuclei in each field of view. Oct4, Sox2, or EdU were manually counted and the total number of positive cells was divided by the total number of cells to determine the percent positive. We used IMARIS software (Oxford Instruments) to generate 3D renderings.

## Traction force microscopy

Polyacrylamide gels of 7.9 kPa stiffness were made by adjusting acrylamide and bisacrylamide stock solution (Bio-Rad Laboratories, Hercules, CA, USA) concentrations. A solution of 40% acrylamide, 2% bisacrylamide, and 1× PBS was polymerized by adding tetramethylethylene diamine

(Fisher BioReagents) and 1% ammonium persulfate. A droplet of the gel solution supplemented with 0.2 µm fluorescent beads solution (Molecular Probe, Fisher Scientific) was deposited on a quartz slide (Fisher Scientific) and covered with a 25 mm glass (Fisher) coverslip pretreated with 3-aminopropyltrimethoxysilane (Sigma-Aldrich) and glutaraldehyde (Sigma-Aldrich). After polymerization, the gel surface attached to the quartz slide was functionalized with Matrigel via polydopamine. The gel was sterilized and stored in 1× PBS before cell seeding. The traction forces exerted by colonies on the polyacrylamide gel substrates were computed by measuring the displacement of fluorescent beads embedded within the gel. Briefly, images of bead motion near the substrate surface, distributed in and around the contact region of a single cell (before and after cell detachment with 10% sodium dodecyl sulfate), were acquired with Yokogawa CSU-21/Zeiss Axiovert 200 M inverted spinning disk microscope with a Zeiss LD C-Apochromat 40×, 1.1 NA water-immersion objective and an Evolve EMCCD camera (Photometrics). The traction stress vector fields were generated using an open-source package of FIJI plugins (https://sites.google.com/site/qingzongtseng/tfm).

## Colony formation assay

To determine clonogenic potential, cells were dissociated with Accutase and plated on Matrigel-coated six-well dishes at a density of 1000 cells per cm (*Nakamura et al., 2016*) in the presence of ROCKi (10 µM). Five days after plating, cells were stained for alkaline phosphatase as per the manufacturer's protocol (StemAb Alkaline Phosphatase Staining Kit II, ReproCell #00-0055) and imaged using a Leica DFC 7000t microscope. To quantify the number of alkaline phosphatase positive colonies, images were analyzed using Fiji (*Schindelin et al., 2012*).

## Library preparation and RNAseq

RNA was extracted using RNeasy Mini kits (QIAGEN) according to the manufacturer's instructions and concentrations were determined by NanoDrop. Library preparation and RNAseq were performed by Novogene Co. Ltd (USA). Briefly, RNA purity was measured using a NanoPhotometer spectrophotometer (IMPLEN). RNA integrity and quantity were determined using a Bioanalyzer 2100 system (Agilent Technologies). Three paired biological replicate libraries were prepared for each condition, with each library generated with 1 µg of RNA per sample. Sequencing libraries were generated using NEBNext Ultra RNA Library Prep Kit for Illumina (NEB) following the manufacturer's recommendations and index codes were added to attribute sequences to each sample. Briefly, mRNA was purified from total RNA using poly-T oligo-attached magnetic beads. Fragmentation was carried out using divalent cations under elevated temperature in NEBNext First Strand Synthesis Reaction Buffer (5×). First strand cDNA was synthesized using random hexamer primer and M-MuLV Reverse Transcriptase (RNase H-). Second strand cDNA synthesis was subsequently performed using DNA Polymerase I and RNase H. Remaining overhangs were converted into blunt ends via exonuclease/polymerase activities. After adenylation of 3′ ends of DNA fragments, NEBNext Adaptor with hairpin loop structure were ligated to prepare for hybridization. In order to select cDNA fragments of preferentially 150–200 bp in length, the library fragments were purified with AMPure XP system (Beckman Coulter, Beverly, MA, USA). Then 3 µL USER Enzyme (NEB, USA) was used with size-selected, adaptor-ligated cDNA at 37°C for 15 min followed by 5 min at 95°C before PCR. Then PCR was performed with Phusion High-Fidelity DNA polymerase, Universal PCR primers, and Index (X) Primer. At last, PCR products were purified (AMPure XP system) and library quality was assessed on the Agilent Bioanalyzer 2100 system.

## RNAseq analysis

Raw data (raw reads) were processed through fastp to remove adapters, poly-N sequences, and reads with low quality. Q20, Q30, and GC content of the clean data were calculated and found to be within the normal range. All the downstream analyses were based on the clean data with high quality. Reference genome (ID: 51) and gene model annotation files were downloaded from genome website browser (NCBI) directly. Paired-end clean reads were aligned to the reference genome using the Spliced Transcripts Alignment to a Reference (STAR) software. FeatureCounts was used to count the read numbers mapped of each gene. And then RPKM of each gene was calculated based on the length of the gene and reads count mapped to this gene. Differential expression analysis was performed using DESeq2 R package. The resulting p values were adjusted using the Benjamini and Hochberg's approach for controlling the FDR. Genes with a padj<0.05 found by DESeq2 were assigned

as differentially expressed. The R package clusterProfiler was used to test the statistical enrichment of differential expression genes in KEGG pathways. KEGG terms with padj<0.05 were considered significant enrichment. Transcription factor binding motif analysis was performed using Enrichr (*Chen et al., 2013*; *Kuleshov et al., 2016*). To investigate YAP target gene expression, supplementary tables generated as previously described (see 'Supplemental Methods' in *Pagliari et al., 2021* and *Estarás et al., 2017*) were used to generate YAP target gene lists (*Estarás et al., 2017*; *Pagliari et al., 2021*).

## Plasmids, site-directed mutagenesis, and generation of lentivirus

pGAMA-YAP was a gift from Miguel Ramalho-Santos (Addgene plasmid #74942). Site-directed mutagenesis was performed on the pGAMA-YAP construct to create p-GAMA-YAP-S127A using the QuikChange Lightning kit (Agilent Technologies #210513). Forward primers used for the Ser-to-Ala substitution were as follows: 5′-GTTCGAGCTCATGCCTCTCCAGC-3′ and 5′-GCTGGAGAGGCATGAGCTCGAAC-3′. The pGAMA-YAP-S127A plasmid was confirmed via DNA sequencing. To prepare lentivirus, HEK293-FT (Invitrogen #R70007) cells were grown in Dulbecco's Modified Eagle Medium (Thermo Fisher #11965118) supplemented with 10% fetal bovine serum (Peak Serum #PS-FB4), non-essential amino acids (UCSF CCF #CCFGA001), pen/strep (UCSF CCF #CCFGK003), and sodium pyruvate (UCSF CCF #CCFGE001) and maintained at 37°C with 5% $CO_2$. Lentivirus was generated according to the manufacturer's specifications by co-transfecting HEK293-FTs with a mixture of packaging plasmids (ViraPower Lentivirus Expression System; Thermo Fisher #K497500). Briefly, $5 \times 10^6$ HEK293-FTs were seeded onto a 10 cm dish containing 10 mL of complete medium without antibiotics. After 24 hr, cells were transfected with a mixture of 3 μg of the lentiviral plasmid containing the gene of interest and 9 μg of the ViraPower Packaging Mix using Lipofectamine 2000 (Thermo Fisher #11668030). At 72 hr post-transfection, supernatant was collected, filtered, and concentrated using Lenti-X Concentrator (Takarabio #631231). Concentrated viral supernatant was aliquoted and stored at –80°C.

To generate hESC lines stably expressing YAP-WT and YAP-S127A, primed HUES8 cells were grown to approximately 60% confluency in one well of a six-well plate. Medium was aspirated, washed once with PBS, and cells were then fed with 1 mL fresh media containing 2 μg of polybrene (Millipore Sigma #TR-1003-G), and incubated for 15 min at 37°C. Concentrated virus supernatant (100 μL) was added and after 6–8 hr 1 mL of fresh medium was added. After 36 hr, viral particles were removed by replacing medium. Three days after virus infection, hESCs were passaged and expanded to three wells in a six-well plate. After wells reached ~75% confluency, hESCs were sorted for high mCherry expression by using a BD FACS Aria3u. Sorted cells were maintained as described above with the addition of penicillin and streptomycin for 3 days, after which cells were then maintained in standard antibiotic-free mTeSR-1 media.

## Western blot analysis

Total lysates were obtained from primed cells, naïve cells, and cells dedifferentiated in the presence of 80 μM CK666 at day 6 which were plated on Matrigel-coated six-well plates. Cells were washed 3× in cold PBS on ice, lysed in 100 μL/well RIPA buffer (50 mM Tris-HCl, 150 mM NaCl, 1 mM EDTA, 1% NP-40, 0.5% deoxycholate, 0.1% SDS supplemented with protease inhibitors PEFA) while rotating at 4°C for 10 min. Cells were then scrapped into microfuge tubes, and a post-nuclear supernatant was collected after centrifugation at 12,000 rpm for 5 min at 4°C. Proteins in lysates were separated by SDS-PAGE and transferred to polyvinylidene difluoride membranes. The membranes were blocked in Tris-buffered saline (TBS) containing 0.1% Tween and 5% non-fat dry milk (TBST) for 60 min at RT and incubated overnight at 4°C with antibodies in TBST containing 5% non-fat dry milk. After washing, membranes were incubated with peroxidase-conjugated secondary antibodies (Jackson ImmunoResearch Laboratories) in TBST for 1 hr at RT, and bound antibodies were developed by enhanced chemiluminescence using SuperSignal West Femto (Thermo Fisher Scientific) and imaged using an Alpha Innotech FluorChem Q (Alpha Innotech). For quantification, the average intensity of each YAP band was normalized to that of the GAPDH band in each sample.

## Quantification and statistical analysis

All statistical tests and sample sizes are included in the Figure Legends and text. All data shown are mean ± SD. In all cases, p values are represented as follows: ****p<0.0001, ***p<0.001, **p<0.001,

and $*p<0.05$. All quantifications were statistically analyzed using ANOVA with Tukey post hoc tests. Statistical analysis was performed using GraphPad Prism version 10.1.2.

## Acknowledgements

The authors would like to acknowledge Dr. Torsten Wittmann and the staff within the Biological Imaging Development CoLab (BIDC) at UCSF Parnassus Heights for their training and support in using the Yokogawa CSU-W1/SoRa super-resolution microscope (NIH Shared Equipment Grant: S10OD028611-01).

## Additional information

### Funding

| Funder | Grant reference number | Author |
| --- | --- | --- |
| National Science Foundation | 1933240 | Nathaniel Paul Meyer<br>Todd G Nystul<br>Diane L Barber |
| National Cancer Institute | 9938488 | Nathaniel Paul Meyer<br>Diane L Barber |
| National Heart, Lung, and Blood Institute | F31HL162520 | Tania Singh |
| Eunice Kennedy Shriver National Institute of Child Health and Human Development | HD055764 | Nathaniel Paul Meyer<br>Todd G Nystul<br>Diane L Barber |
| National Institute of General Medical Sciences | GM136348 | Nathaniel Paul Meyer<br>Todd G Nystul |

The funders had no role in study design, data collection and interpretation, or the decision to submit the work for publication.

### Author contributions

Nathaniel Paul Meyer, Conceptualization, Data curation, Formal analysis, Validation, Visualization, Methodology, Writing - original draft, Project administration, Writing - review and editing; Tania Singh, Software, Investigation, Writing - review and editing; Matthew L Kutys, Formal analysis, Writing - review and editing; Todd G Nystul, Resources, Software, Supervision, Funding acquisition, Project administration, Writing - review and editing; Diane L Barber, Conceptualization, Formal analysis, Supervision, Funding acquisition, Validation, Investigation, Methodology, Project administration, Writing - review and editing

### Author ORCIDs

Nathaniel Paul Meyer http://orcid.org/0000-0002-8327-8563
Tania Singh https://orcid.org/0000-0003-0692-4821
Matthew L Kutys https://orcid.org/0000-0002-0752-649X
Todd G Nystul https://orcid.org/0000-0002-6250-2394
Diane L Barber https://orcid.org/0000-0001-7185-9435

### Decision letter and Author response

Decision letter https://doi.org/10.7554/eLife.89725.sa1
Author response https://doi.org/10.7554/eLife.89725.sa2

## Additional files

### Supplementary files
• MDAR checklist

## Data availability

RNA sequencing data generated for this manuscript were deposited in NCBI Gene Expression Omnibus (accession number: GSE276968). All data generated or analyzed during this study are included in the manuscript or supporting files.

The following dataset was generated:

| Author(s) | Year | Dataset title | Dataset URL | Database and Identifier |
|---|---|---|---|---|
| Meyer NP, Singh T, Kutys ML, Nystul T, Barber D | 2024 | Arp2/3 Complex Activity Enables Nuclear YAP for Naïve Pluripotency of Human Embryonic Stem Cells | https://www.ncbi.nlm.nih.gov/geo/query/acc.cgi?acc=GSE276968 | NCBI Gene Expression Omnibus, GSE276968 |

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
