## [Editor Report]

This important work identifies a mechanism for cytoskeletal dynamics coordinating cell mechanics to regulate gene expression and facilitate transitions between pluripotency states in human embryonic stem cells. The data were collected and analyzed using convincing and validated methodology and can be used as a starting point for studies of the cell biology of embryonic stem cells.

---

## [Decision Letter]

[Editors' note: this paper was reviewed by Review Commons.]

---

## [Author Response]

General Statements [optional]

We thank the reviewers for their comments and constructive suggestions. We are submitting a revised manuscript and supplemental materials with new data and edits that we believe effectively address reviewers’ requests and suggestions.

Point-by-point description of the revisions

The results of experiments where Arp2/3 is blocked (Figure 2) should be confirmed by Arp2/3 knock-down and with an independent Arp2/3 inhibitor. Several are available (CK-869, Benproperine, Pimozide). For Figure 3 and 4, that would not be necessary, but to establish the specificity of the effect in Figure 2 this is absolutely required.

As requested, we include new data with CK-869, the suggested additional Arp2/3 complex inhibitor. These new data, included in Figure S2A-C, confirm that CK-869, like our previous findings with CK-666, blocks assembly of the actin filament ring around naïve colonies of hESCs without effects on pluripotency, indicated by the naïve markers Oct4 and *Sox2*, or on cell proliferation, indicated by EdU incorporation.

To our knowledge, there is only one publication (PMID: 31571309) that Pimozide binds to ARPC2 and inhibits Arp2/3 complex activity. It is more widely known as an antipsychotic by antagonizing dopamine and 5-HT receptors. Hence, its selectivity for inhibiting Arp2/3 complex activity is questionable. Additionally, to our knowledge, there are only 2 publications (PMID: 36558913; PMID: 30710516) that Benproperine inhibits ARPC2, compared with many publications on its antitussive effects. Hence, like Pimozide, its selectivity for inhibiting Arp2/3 complex activity is not well established.

We respectfully disagree with generating a Arp2/3 knock-down hESC line. Arp2/3 complex genes are known to be essential in both mouse and human embryonic stem cells (PMID: 29662178 and PMID: 31649057). Furthermore, reports on successful knockout of complex subunits indicate that additional genetic manipulations are needed to maintain cell survival, including knockout of INK4A/ARF to bypass apoptosis associated with Arp2 shRNA knockdown (PMID: 22385962) and genetic manipulations in mouse models (PMID: 22492726). Thus, knock-down of Arp2/3 complex is not reasonable for our study nor we believe necessary based on our new data with CK-869.

I believe that the status of the actin cytoskeleton in both states is not well enough characterized. This is especially obvious for branched actin networks themselves that depend on the Arp2/3. To this end, the authors may localize Arp2/3 or cortactin, a useful surrogate that often gives a better staining. This point is particularly important since contractile fibers are not made of branched actin. Myosin cannot walk or pull along branched actin networks because of steric hindrance. It might well be that branched actin networks are debranched after Arp2/3 polymerization. I suggest staining tropomyosins that would indicate where the transition between branched and unbranched actin would be. Along this line, phosphoERMs should be localized and revealed by Western blots (we expect an increase from primed to naive state) because they cannot perform the proposed function of linker between the membrane and actin filaments if they are not phosphorylated.

As requested, we include new data with cortactin immunolabeling, which are shown in Figure 1F and Figure S2D. These new data confirm that cortactin immunolabeling overlaps with the actin filament ring around naïve hESC colonies (Figure 1F) and its localization is disrupted by inhibiting Arp2/3 complex activity with CK-666 (Figure S2D). These new data and our previous data with pMLC immunolabeling suggest that the actin ring is composed of both branched and unbranched filaments, and we added this comment in our Results section.

Also as requested, we immunoblotted cell lysates for phosphorylated ERMs; however, we did not see changes in naïve compared with primed hESCs. With the new requested data on cortactin localization, which we believe and as the Reviewer indicates is a better indicator of actin architecture in the ring, we omitted our previous data on ERM binding in Figure 1.

Branched actin is required for cell cycle progression and cell proliferation in normal cells. This requirement is lost in most cancer cells (Wu et al., Cell 2012; Molinie et al., Cell Res 2019). This would be really important to know whether ESCs stop proliferating upon CK-666 treatment. In other words, do they behave like normal cells or transformed cells. Proliferation is a major function that depends on the YAP pathway. Cell counts and EdU incorporation can easily provide answers to this important question.

As requested, we include new data on proliferation with Edu incorporation as indicated by Reviewer 1. These new data, shown in Figure S2C, indicate proliferation is not changed by either CK-666 or CK-869 compared with untreated controls.

Minor comments 4-6 and 8.

As requested, we made appropriate corrections.

Minor comment 7. What about the rescue of cell morphology? Does active YAP restore the intercellular contractile bundle?

As requested, we include new data in Figure 4 indicating that heterologous expression of active nuclear localized YAP-S127A but not YAP-WT does restore formation of the actin ring in the presence of CK666. Accordingly, we added text changes in our Results and Discussion section suggesting a reciprocal signaling circuit of actin filament remodeling being both upstream and downstream of active nuclear-localized YAP.

The authors found that a ring of actin filaments at the colony periphery was characteristic of the naive hESCs. However, because all the data are presented as an image of a single confocal section, the 3D organization of the actin filaments is not clear. Although the authors drew a scheme for this actin ring being located in the apical domain of polarized cells, such data have not been provided in the manuscript. Since naive hESCs form dome-like colonies, it is important to show the 3D organization of actin filaments in the colony. 3D reconstruction of confocal microscopy images of the naive hESC colonies is required to show the relationship between actin filaments, adherens junctions, and the nuclei (as a reference for the Z axis). If 3D reconstruction is not technically possible, confocal images at different Z levels and maximum projection images should be obtained and provided.

As requested, our revision includes new 3D images (Figure S1E) of the actin ring generated by using Imaris software (Oxford Instruments), which we previously used to show 3D images of mitochondrial morphology (PMID: 34038242). We found this analysis demonstrated the difficulty in determining exact positional organization in regard to the actin filaments, adherens junctions, and nuclei. From these new data we are confident in our conclusion that the actin fence surrounds naïve colonies and is present in cells at the colony periphery. We agree with the reviewer that our previous statements on the 3D organization and positional information regarding the actin fence is insufficiently demonstrated by our current data to be confident in those conclusions. As such we have omitted the scheme, which implied a specific 3D organization, and have removed any comments on the relative organization aside from its location in cells on naïve colony boarders from the manuscript.

Some of the statistical analyses were inappropriate. The authors have used Student's t-test for all analyses; however, one-way ANOVA and post-hoc analysis must be used to compare three or more groups (Figures 2B, D, E, 3G, 4B, D, E).

As requested, all relevant statistical analyses have been repeated using ANOVA and post-hoc analysis. Figure legends and methods have been updated to reflect this where appropriate.

Minor Comments 3.Page 9, second paragraph. In the Discussion section, authors have written that "Cells within the ICM of mouse blastocysts exclude YAP from the nucleus whereas cells within the ICM of human blastocysts maintain nuclear YAP." However, a recent study has reported that the ICM/epiblast of mouse late blastocysts also express nuclear YAP. Epiblast Formation by TEAD-YAP-Dependent Expression of Pluripotency Factors and Competitive Elimination of Unspecified Cells. Hashimoto M, Sasaki H. Dev Cell. 2019, 50:139-154.e5. doi: 10.1016/j.devcel.2019.05.024.

We thank the reviewer for alerting us to this publication. Results in this publication are in agreement with the statement we made in our Discussion section regarding the cellular localization of YAP in murine blastocysts at the relevant time point. Naïve stem cells are isolated from murine blastocysts at E3.5. This publication, as well as previous publications from this group (Hirate et al., 2012, Nishioka et al., 2009), find that YAP is excluded from cells within the ICM at this time point. In the citation provided by the reviewer, Figure 2 provides confirmation of previous data showing that cells within the ICM of E3.5 mouse blastocysts exclude YAP from the nucleus.

We note, however, that fundamental misunderstandings of the literature occur based on differences in cell models. Mouse naïve stem cells are isolated from the ICM of murine embryos at E3.5 and are distinctly different that clonal hESCs used in our study. The publication indicated by the reviewer, when carefully evaluated, agrees with our findings and conclusions in indicating “However, we previously showed that in the ICM of early blastocyst stage embryos, active Hippo signaling inactivated TEAD proteins by excluding YAP from the nuclei (Hirate et al., 2012, Nishioka et al., 2009).” Their final figure (Figure 7) also shows this in a diagram with E3.5 of YAP excluded from the nucleus. These E3.5 cells are where mouse naïve stem cells come from, and thus represent to the best of our knowledge the most appropriate comparison with the cells we used in our study.

Many of their conclusions seem to be based on the qualitative analysis of a single image (e.g. Figures 1D-G, Figure 2G, Supplementary Figure 2). The authors should provide quantitative information regarding these analyses and indicate the number of cells/replicas collected for each experiment.

As requested, our revision includes quantitative data where feasible. We now include additional quantification of *Sox2*/Oct4 double positive cells to ensure pluripotency and quantification of EdU+ cells for assessing proliferation. Our data demonstrating co-localization of pMLC and βcatenin are from at least 3 separate cell preparations, as we indicate. Additionally, in the field, traction force microscopy is not commonly quantified beyond including scale bars, which our original manuscript shows. These experiments were also completed in triplicate, with representative data shown.

Many of the images seem to require a flat-field correction. Could the authors check that the illumination is homogeneous? This artifact could affect the data analysis.

In response, the spinning disk microscopes we used for all images are equipped with a Borealis that provides uniformity of illumination.

The actin ring surrounding hESCs colonies was previously described by Närvä et al. Although the authors cited this previous work, they do not discuss in deep the differences and similarities with their observations.

As requested, we added additional comments relative to the findings in Närvä et al., which was included as a citation in our original manuscript. These new comments in the Results and Discussion section describe differences between our findings and those of Närvä et al., including that we observe an actin filament ring only in the naïve state of pluripotency, whereas Närvä et al. observe a related actin architecture in the primed state. However, Närvä et al. use induced pluripotent stem cells, which are proposed to be closer to naïve pluripotency than primed stem cells as conventionally isolated and maintained (see PMID: 27424783 and PMID: 19497275). Additionally, we observe that the contractile actin ring in naïve pluripotent stem cells is in a higher z-plane than reported by Närvä et al., although a direct comparison is difficult to make. Moreover, we retain our original comparison, which indicated “A similar actin ring is reported to encircle colonies of clonal human pluripotent stem cells to provide a mechanosensitive element linked to focal adhesions (Närvä citation). The actin filament ring we observed around naïve hESC colonies was instead tethered to adherens junctions, as indicated by co-labeling for β-catenin.” Our findings, however, are in agreement with those in Närvä et al., in showing that a colony of pluripotent stem cells exist as a cohesive unit with a contractile actin architecture at the colony periphery with a less mechanical interior. Although Närvä et al., do not link their actin phenotype to functional pluripotency, they do demonstrate that interference with mechanosensing proteins within hESCs alters the state of pluripotency and subsequent differentiation potential.

There are many experimental details missing that are extremely relevant to fully understand the experiments and evaluate the robustness of the analyses (e.g., microscopy setup, fluorescent probes used for immunostaining, incubation conditions with the inhibitors SMIFH2 and CK666).

As requested, our revision addresses this comment by including additions when indicated in the Methods.

The qualitative observation of Figure 3F suggests a lower overall YAP levels in primed and +CK666 cells in comparison to naive cells. Could the authors check if this is correct and, if this is the case, explain the observation?

As requested, we include new data on immunoblotting for total YAP in lysates from primed, naïve, and +CK666 treated cells in Supplementary Figure 3. These indicate that YAP abundance is largely the same, with some variability in the total amount of YAP in +CK666 treated cells. We conclude from these data that the importance of YAP in naïve pluripotency is its cellular localization and not total abundance.

The authors should discuss deeper the rationale of the pan-ERM immunostaining experiments (since they used the individual antibodies afterwards) and provide a brief discussion of their results and, in particular, the colocalization with moesin but not with ezrin or radixin.

As indicated in our response to Review 1 comment 2, data on ERM immunolabeling is omitted in our revision in lieu of requested new data with cortactin immunolabeling.

The Introduction makes the reader think that actin is the only cytoskeletal network involved in embryo development and stem cell properties. They should also include a brief discussion on the relevance of the other cytoskeletal networks in mechanotransduction and cell fate decisions.

As requested, our revised Introduction now includes the following:

“Other components of the cytoskeleton such as microtubules and intermediate filaments have also been established to modulate stem cell behavior although studies have primarily focused recently on how they impact nucleus morphology and activity^19–21^.”

There are many abbreviations that are not defined in the text and are extremely specific to the field.

As requested, where appropriate for the field, abbreviations have been defined in our revision.

Could the authors explain the selection of the pluripotency markers studied by qPCR? Specifically, why they studied DNMT3L, DPPA3, KLF2, and KLF4 (Figure 1B) and the different set PECAM1, ESRRB, KLF4, and DNMT3L in Figure 2B.

We selected naïve pluripotency markers based on a number of sources studying the transcriptional regulation and differences in human naïve pluripotency (PMID: 28429706, PMID: 30673604, PMID: 29129686, and PMID: 37106060). Broadly speaking, we chose markers consistent across multiple reports as markers of human naïve pluripotency. Additionally, we chose markers of interest to many in the field, such as ESRRB, which has been reported to facilitate transition of human naïve pluripotent stem cells through a state of pluripotency called formative pluripotency. Additionally, we chose markers such as DPPA3 and DNMT3L due to their role in epigenetics, which is also of relevant interest to the field. Lastly, KLF2 and KLF4 are classical markers of naïve pluripotency and have been well established to facilitate that state. To facilitate comparison, we have adjusted Figure 1B and Figure 2B to use the same markers.

Figures 1G and 2G, please include the images of the colonies.

As requested, phase contrast images are now included for Fig1G in Figure S1D and for Figure 2G in Figure S2E.